# Interpolating Between Sampling and Variational Inference with Infinite Stochastic Mixtures

**Richard D. Lange**[1]     **Ari S. Benjamin**[1]     **Ralf M. Haefner**[*2]     **Xaq Pitkow**[*3]

[1]Dept. of Neurobiology, University of Pennsylvania, Philadelphia, Pennsylvania, USA
[2]Dept. of Brain and Cognitive Sciences, University of Rochester, Rochester, New York, USA
[3]Baylor College of Medicine, Rice University, Houston, Texas, USA
[*]equal contribution

## Abstract

Sampling and Variational Inference (VI) are two large families of methods for approximate inference that have complementary strengths. Sampling methods excel at approximating arbitrary probability distributions, but can be inefficient. VI methods are efficient, but may misrepresent the true distribution. Here, we develop a general framework where approximations are stochastic mixtures of simple component distributions. Both sampling and VI can be seen as special cases: in sampling, each mixture component is a delta-function and is chosen stochastically, while in standard VI a single component is chosen to minimize divergence. We derive a practical method that interpolates between sampling and VI by analytically solving an optimization problem over a mixing distribution. Intermediate inference methods then arise by varying a single parameter. Our method provably improves on sampling (reducing variance) and on VI (reducing bias+variance despite increasing variance). We demonstrate our method's bias/variance trade-off in practice on reference problems, and we compare outcomes to commonly used sampling and VI methods. This work takes a step towards a highly flexible yet simple family of inference methods that combines the complementary strengths of sampling and VI.

## 1 INTRODUCTION

We are concerned with the familiar and general case of approximating a probability distribution, such as occurs in Bayesian inference when both the prior over latent variables and the likelihood function connecting them to data are known, but computing the posterior exactly is intractable. There are two largely separate families of techniques for approximating such intractable inference problems: Markov Chain Monte Carlo (MCMC) sampling, and Variational Inference (VI) [Bishop, 2006, Murphy, 2012].

Sampling-based methods, including MCMC, approximate a distribution with a finite set of representative points. MCMC methods are stochastic and sequential, generating a sequence of sample points that, given enough time, become representative of the underlying distribution increasingly well. MCMC sampling is (typically) asymptotically unbiased, at the expense of high variance, leading to long run times in practice. Similar to the approach we take here, sampling methods are studied at different scales: both in terms of their asymptotic limit (i.e. their bias at infinitely many samples) and their practical behavior for finite samples or other resource limits [Korattikara et al., 2014, Angelino et al., 2016].

Variational Inference (VI) refers to methods that produce an approximate distribution by minimizing some quantification of divergence between the approximation and the desired posterior distribution [Blei et al., 2017, Zhang et al., 2019]. For the purposes of this paper, we will use VI to refer to the most common flavor of variational methods, namely minimizing the Kullback-Leibler (KL) divergence between an approximate distribution from a fixed family and the desired distribution [Bishop, 2006, Wainwright and Jordan, 2008, Murphy, 2012, Blei et al., 2017]. The best-fitting approximate distribution is often used directly as a proxy for the true posterior in subsequent calculations, which can greatly simplify those downstream calculations if the approximate distribution is itself easy to integrate. In contrast to MCMC, VI is often used in cases where speed is more important than asymptotic bias [Angelino et al., 2016, Blei et al., 2017, Zhang et al., 2019].

Our goal is to develop an intermediate family of methods that "interpolate" between MCMC and VI, inspired by a simple and intuitive picture (Figure 1): we propose applying sampling methods *in the space of variational parameters* such that the resulting approximation is a stochastic mixture

*Accepted for the 38th Conference on Uncertainty in Artificial Intelligence* (UAI 2022).

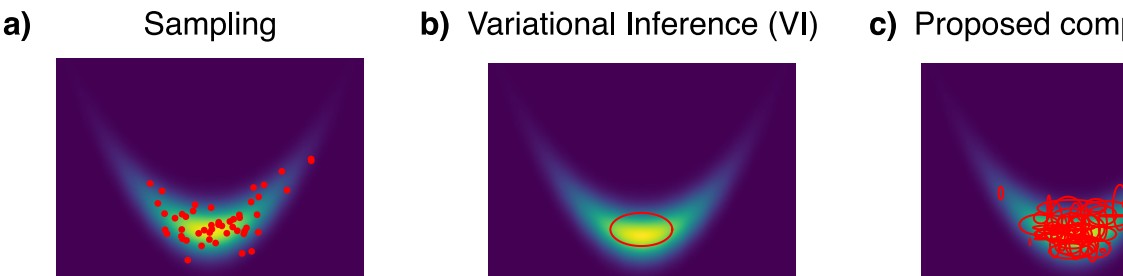

| a) Sampling | b) Variational Inference (VI) | c) Proposed compromise |

Figure 1: Conceptual introduction on a simple 2D example – the "banana" distribution. **a)** Sampling methods approximate the underlying $p(\mathbf{x})$ with a stochastic set of representative points, $\mathbf{x} \sim p(\mathbf{x})$. **b)** Variational Inference (VI) methods begin by selecting an approximating distribution family, $q(\mathbf{x}; \theta)$, here a Gaussian with diagonal covariance plotted as an ellipse at its $1\sigma$ contour. The optimal parameters $\theta^*$ are chosen to minimize $\mathrm{KL}(q(\mathbf{x}; \theta)||p(\mathbf{x}))$. **c)** We propose using a stochastic mixture of component distributions, where *parameters $\theta$* are sampled from a "mixing distribution" $\psi(\theta)$, i.e. $\theta \sim \psi(\theta)$.

of variational "component" distributions [Yin and Zhou, 2018]. This extends sampling by replacing the sampled points with extended components, and it extends VI by replacing the single best-fitting variational distribution with a stochastic mixture of more localized components. This is qualitatively distinct from previous variational methods that use *stochastic optimization*: rather than stochastically optimizing a single variational approximation [Hoffman et al., 2013, Salimans et al., 2015], we use stochasticity to construct a *random mixture* of variational components that achieves lower asymptotic bias than any one component could. As we will show below, this framework generalizes both sampling and VI, where sampling and VI emerge as special cases of a single optimization problem.

This paper is organized as follows. In section 2, we set up the problem and our notation, and describe how both classic sampling and classic VI can be understood as special cases of stochastic mixtures. In section 3, we introduce an intuitive framework for reasoning about infinite stochastic mixtures and define an optimization problem that captures the trade-off between sampling and VI. Section 4 introduces an approximate objective and closed-form solution and describes a simple practical algorithm. Section 5 gives empirical and theoretical results that show how our method interpolates the bias and variance of sampling and VI. Finally, section 6 concludes with a summary, related work, limitations, and future directions.

## 2  SETUP AND NOTATION

Let $p^*(\mathbf{x}) = Zp(\mathbf{x})$ denote the unnormalized probability distribution of interest, with unknown normalizing constant Z. For instance, in the common case of a probabilistic model with latent variables $\mathbf{x}$, observed data $\mathcal{D}$, and joint distribution $p(\mathbf{x}, \mathcal{D})$, we are interested in approximations to the posterior distribution $p(\mathbf{x}|\mathcal{D})$. This is intractable in general,

but we assume that we have access to the un-normalized posterior $p^*(\mathbf{x}|\mathcal{D}) = \frac{1}{Z}p(\mathcal{D}|\mathbf{x})p(\mathbf{x})$.[1] Let $q(\mathbf{x}; \theta)$ be any "simple" distribution that may be used used in a classic VI context (such as mean-field or Gaussian), and let $m_T(\mathbf{x})$ be a mixture containing $T$ of these simple distributions as components, defined by a set of $T$ parameters $\{\theta^{(1)}, \ldots, \theta^{(T)}\}$:

$$m_T(\mathbf{x}) \equiv \frac{1}{T}\sum_{t=1}^{T} q(\mathbf{x}; \theta^{(t)}). \tag{1}$$

For example, if q is a multivariate normal with mean $\mu$ and covariance $\Sigma$, then $\theta^{(t)} = \{\mu^{(t)}, \Sigma^{(t)}\}$ and $m_T(\mathbf{x})$ would be a mixture of $T$ component normal distributions [Gershman et al., 2012, Zobay, 2014].

We will study properties of mixing distributions, or distributions over component parameters, which we denote $\psi(\theta)$ [Ranganath et al., 2016]. If the set of $\theta^{(t)}$ is drawn randomly from $\psi(\theta)$, then as $T \to \infty$, $m_T(\mathbf{x})$ approaches the idealized infinite mixture,

$$m(\mathbf{x}) \equiv \int_\theta q(\mathbf{x}; \theta)\psi(\theta)d\theta. \tag{2}$$

**Sampling and VI as special cases of the mixing distribution.**  Let $\theta^* = \arg\min_\theta \mathrm{KL}(q(\mathbf{x}; \theta)||p(\mathbf{x}))$ be the parameters corresponding to the classic single-component variational solution. VI corresponds to the special case where the mixing distribution $\psi(\theta)$ is a Dirac delta around $\theta^*$, or $\psi(\theta) = \delta(\theta - \theta^*)$, in which case the mixture $m_T(\mathbf{x})$ is equivalent to $q(\mathbf{x}; \theta^*)$ regardless of the number of components $T$. Sampling can also be seen as a special case of $\psi(\theta)$ in which each component narrows to a Dirac delta ($\psi(\theta)$ places negligible mass on regions of $\theta$-space where components have appreciable width), and the means of the components are distributed according to $p(\mathbf{x})$. This requires

---

[1] To reduce clutter, $\mathcal{D}$ will be dropped in the remainder of the paper, and we will use only $p(\mathbf{x})$ and $p^*(\mathbf{x})$.

that the component family $q(\mathbf{x}; \theta)$ is capable of expressing a Dirac-delta at any point $\mathbf{x}$, such as a location-scale family. Thus, both sampling and VI can be seen as limiting cases of stochastic mixture distributions, $m_T(\mathbf{x})$, defined by a distribution over component parameters, $\psi(\theta)$. In what follows, we will show how designing the mixing distribution $\psi(\theta)$ allows us to create mixtures that trade-off the complementary strengths of sampling and VI.

# 3 CONCEPTUAL FRAMEWORK

## 3.1 DECOMPOSING $\mathrm{KL}(m||p)$ INTO MUTUAL INFORMATION AND EXPECTED KL

The idealized infinite mixture $m(\mathbf{x})$ is fully defined by the chosen component family $q(\mathbf{x}; \theta)$ and the mixing distribution $\psi(\theta)$. Consider the variational objective with respect to the entire mixture, $\mathrm{KL}(m||p)$:

$$\mathrm{KL}(m||p) = \int_{\mathbf{x}} m(\mathbf{x}) \log \frac{m(\mathbf{x})}{p^*(\mathbf{x})} d\mathbf{x} + \log Z, \quad (3)$$

where $Z$ is the normalizing constant of $p^*(\mathbf{x})$ and is irrelevant for constructing $m(\mathbf{x})$. Instead of (3), one can use the equivalent objective of maximizing the *Evidence Lower BOund* or ELBO [Bishop, 2006, Murphy, 2012, Blei et al., 2017]. Regardless, minimizing (3) or maximizing the ELBO for mixtures is intractable in general. However, as first shown by Jaakkola and Jordan [1998] for finite mixtures, it admits the following useful decomposition:

$$\mathrm{KL}(m||p) = \underbrace{\int_{\theta} \psi(\theta) \int_{\mathbf{x}} q(\mathbf{x}; \theta) \log \frac{q(\mathbf{x}; \theta)}{p^*(\mathbf{x})} d\mathbf{x}\, d\theta}_{\text{(i) Expected KL}}$$
$$- \underbrace{\int_{\theta} \psi(\theta) \int_{\mathbf{x}} q(\mathbf{x}; \theta) \log \frac{q(\mathbf{x}; \theta)}{m(\mathbf{x})} d\mathbf{x}\, d\theta}_{\text{(ii) Mutual Information } \mathcal{I}[\mathbf{x}; \theta]} \quad (4)$$

(dropping $\log Z$). The first term, (i), is the **Expected KL Divergence** for each component when the parameters are drawn from $\psi(\theta)$. This term quantifies, on average, how well the mixture components match the target distribution. In isolation, Expected KL is minimized when all components individually minimize $\mathrm{KL}(q||p)$, i.e. when $\psi(\theta) \to \delta(\theta - \theta^*)$. This tendency to concentrate $\psi(\theta)$ to the single best variational solution is balanced by the second term, (ii), which is the **Mutual Information** between $\mathbf{x}$ and $\theta$, which we will write $\mathcal{I}[\mathbf{x}; \theta]$, under the joint distribution $q(\mathbf{x}; \theta)\psi(\theta)$. This term should be *maximized*, and, importantly, it does not depend on $p^*(\mathbf{x})$. Mutual Information is maximized when the components are as diverse as possible, which encourages the components to become narrow and to spread out over diverse regions of $\mathbf{x}$ *regardless* of how well they agree with $p(\mathbf{x})$. This decomposition of $\mathrm{KL}(m||p)$ into Mutual Information (between $\mathbf{x}$ and $\theta$) and Expected KL (between

q and p) is convenient because approximations to Mutual Information are well-studied, and minimizing Expected KL can leverage standard tools from VI.

## 3.2 TRADING OFF BETWEEN MUTUAL INFORMATION AND EXPECTED KL

We will refer back to this decomposition of the $\mathrm{KL}(m||p)$ objective into Expected KL and Mutual Information throughout. Figure 2 depicts a two-dimensional space with Expected KL on the x-axis and Mutual Information on the y-axis. Any given mixing distribution $\psi(\theta)$ can be placed as a point in this space, but in general many $\psi(\theta)$'s may map to the same point.

Sampling and VI live at extreme points in this space. Classic VI, where $\psi(\theta) = \delta(\theta - \theta^*)$, corresponds to the blue point (c), because by definition $\theta^*$ achieves the minimum possible KL, and $\mathcal{I}[\mathbf{x}; \theta]$ is zero. Classic sampling corresponds to the green point (d), with $\psi(\theta)$ placing mass only on Dirac-delta-like components, and selecting each component with probability $p(\mu)$, where $\mu$ is the mean of q determined by $\theta$.

Towards the goal of constructing mixtures that trade-off properties of sampling and VI, we propose to view the two terms in (4) as separate objectives that may be differently weighted, and maximizing the objective

$$\mathcal{L}(\psi, \lambda) = \mathcal{I}[\mathbf{x}; \theta] - \lambda \mathbb{E}_\psi [\mathrm{KL}(q||p)] \quad (5)$$

for a given hyperparameter $\lambda$ with respect to the mixing distribution $\psi(\theta)$. This objective may alternatively be viewed as the Lagrangian of a constrained optimization problem over the mixing density $\psi(\theta)$, where Mutual Information is maximized subject to a constraint on Expected KL. This is a concave maximization problem with linear constraints, defining a Pareto front of solutions that each achieve a different balance between Expected KL and Mutual Information. Maximizing Mutual Information necessitates approximations [Poole et al., 2019], so there may be good mixture approximations that are not found in practice, such as the yellow point (e) in Figure 2. In section 4 below, we use an approximation to Mutual Information that has the property, illustrated by the orange curve (f) in Figure 2, of connecting VI (c) to sampling (d), controlled by varying $\lambda$. As shown on the right of Figure 2, our method produces mixtures that behave like classic samples when $\lambda = 1$, that behave like classic VI when $\lambda \to \infty$, and that exhibit intermediate behavior at intermediate values of $\lambda$.

We emphasize that this frame is quite general: any stochastic mixture can be reasoned about in terms of its Expected KL and Mutual Information, and this is a natural space in which to think about interpolating sampling and VI. A similar decomposition of $\mathrm{KL}(m||p)$ (or the ELBO) has been used by previous methods that optimize mixtures [Zobay, 2014, Jaakkola and Jordan, 1998, Gershman et al., 2012,

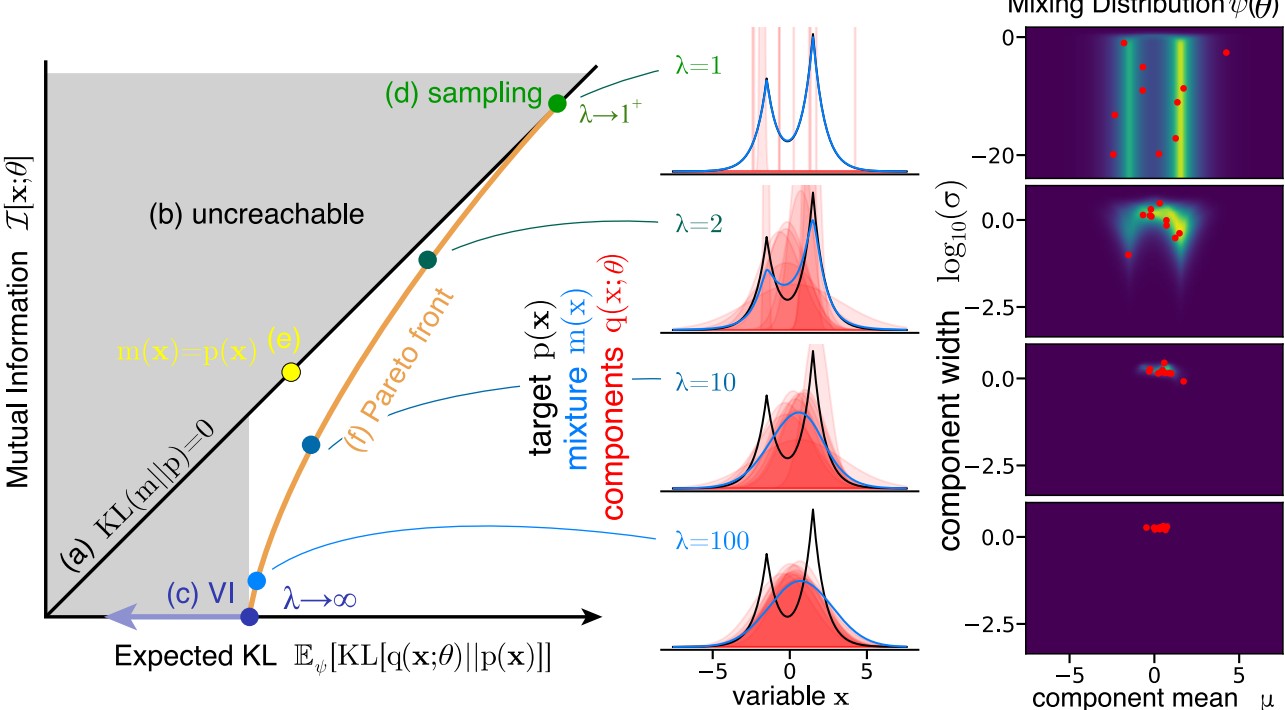

Figure 2: *Left*: Understanding mixtures in terms of Mutual Information and Expected KL. **a)** The quality of any infinite mixture (in terms of KL(m||p)) is given by its distance from the y=x line (black diagonal line). **b)** Two unreachable regions are shaded in gray: above the y=x line (because KL(m||p) $\geq$ 0), and to the left of the single-component variational solution, since VI achieves the minimum KL(q||p). **c)** When $\psi(\theta) = \delta(\theta - \theta^*)$ as in classic VI, Expected KL is at its minimum and Mutual Information is zero. Increasing the expressiveness of q corresponds to moving left along the x-axis (blue arrow). **d)** Because sampling is unbiased, it is a mixture that lives on the KL(m||p) $= 0$ or $y = x$ line. If **x** is discrete, the coordinates of the point marked (d) are $(\mathcal{H}[\mathbf{x}], \mathcal{H}[\mathbf{x}])$, i.e. the entropy of p(**x**). When **x** is continuous, both Mutual Information and Expected KL grow unboundedly together as the individual components narrow. **e)** Any point on the y=x line implies m(**x**) = p(**x**), and this may be possible without resorting to sampling for certain combinations of p and q. However, such mixtures are not guaranteed to exist for all problems, and are difficult to find due to the intractability of Mutual Information. **f)** We propose a family of mixture approximations, parameterized by $\lambda$, that connects VI to sampling in a natural and principled way. Points on this curve correspond to solutions to the (approximate version of the) objective in (5). *Middle*: Examples in a 1D toy problem, where p(**x**) is an unequal mixture of two heavy-tailed distributions (black lines), and q(**x**; $\theta$) is a single Gaussian component with parameters $\theta = \{\mu, \log \sigma\}$ (translucent red components). *Right*: Varying $\lambda$ controls the mixing distribution over $\theta$ (image). Red points correspond to the Gaussian components in the middle.

Yin and Zhou, 2018]. The primary difference between these previous methods is how they approximate (or lower-bound) Mutual Information. In the next section, we introduce a new approximation that is particularly efficient, and is the first to our knowledge that replicates sampling-like behavior with finitely many components.

## 4 APPROXIMATE OBJECTIVE

Maximizing Mutual Information, as is required by (5), is a notoriously difficult problem that arises in many domains, and there is a large collection of approximations and bounds in the literature [Jaakkola and Jordan, 1998, Brunel and Nadal, 1998, Gershman et al., 2012, Wei and Stocker, 2016,

Kolchinsky and Tracey, 2017, Poole et al., 2019]. Previous work has optimized *finite* mixtures by considering how each of $T$ components interacts with the other $T - 1$ components, resulting in quadratic scaling with $T$ [Gershman et al., 2012, Zobay, 2014, Guo et al., 2016, Miller et al., 2017, Kolchinsky and Tracey, 2017, Nalisnick and Smyth, 2017, Yin and Zhou, 2018, Poole et al., 2019]. Beginning instead with *infinite* mixtures, we find that the local geometry of $\theta$-space is sufficient to provide an approximation to Mutual Information *that can be evaluated independently for each value of $\theta$*, leading to linear scaling with $T$.

## 4.1 STAM'S INEQUALITY

Mutual Information between $\mathbf{x}$ and $\theta$ can be written as

$$
\begin{aligned}
\mathcal{I}[\mathbf{x};\theta] &= \mathcal{H}[\theta] - \mathbb{E}_{\mathrm{m}(\mathbf{x})}\left[\mathcal{H}[\hat{\theta}|\mathbf{x}]\right] \\
&= \mathcal{H}[\theta] - \mathbb{E}_{\psi(\theta)}\Big[\underbrace{\mathbb{E}_{\mathrm{q}(\mathbf{x}|\theta)}[\mathcal{H}[\hat{\theta}|\mathbf{x}]]}_{\mathcal{H}[\hat{\theta}|\theta]}\Big]
\end{aligned} \quad (6)
$$

where $\mathcal{H}[\theta]$ is the entropy of $\psi(\theta)$ and $\mathcal{H}[\hat{\theta}|\mathbf{x}]$ is the entropy of $\mathrm{q}(\hat{\theta}|\mathbf{x}) = \frac{\mathrm{q}(\mathbf{x};\hat{\theta})\psi(\hat{\theta})}{\mathrm{m}(\mathbf{x})}$, i.e. the distribution of *inferred* $\theta$ values for a given $\mathbf{x}$. The second line follows simply from expanding the definition of $\mathrm{m}(\mathbf{x})$ in the outer expectation. The term $\mathcal{H}[\hat{\theta}|\theta]$ can be thought of in terms of a statistical estimation problem: $\hat{\theta}$ is the "recovered" value of $\theta$ after passing through the "channel" $\mathbf{x}$. Bounding the error of such estimators is a well-studied problem in statistics.

From (6), a lower-bound on Mutual Information can be derived from an *upper bound* on $\mathcal{H}[\hat{\theta}|\theta]$ for each $\theta$. For this, we draw inspiration from Stam's inequality [Stam, 1959, Dembo et al., 1991, Wei and Stocker, 2016], which states

$$
\mathcal{H}[\hat{\theta}|\theta] \le \frac{1}{2}\log\left|2\pi e \mathcal{F}(\theta)^{-1}\right|, \quad (7)
$$

where $|\cdot|$ is a determinant, and $\mathcal{F}(\theta)$ is the Fisher Information Matrix, defined as

$$
\mathcal{F}(\theta)_{ij} = -\mathbb{E}_{\mathrm{q}(\mathbf{x};\theta)}\left[\frac{\partial^2}{\partial\theta_i\partial\theta_j}\log\mathrm{q}(\mathbf{x};\theta)\right].
$$

The Fisher Information Matrix is also the local metric on the *statistical manifold* with coordinates $\theta$ [Amari, 2016]; it is used to quantify how "distinguishable" $\theta$ is from $\theta + d\theta$. Note that (7) can be viewed as the entropy of a Gaussian approximation to $\mathrm{q}(\hat{\theta}|\mathbf{x})$ with precision matrix $\mathcal{F}(\theta)$; this approximation is most accurate when $\mathrm{q}(\mathbf{x};\theta)$ itself is narrow and approximately Gaussian [Wei and Stocker, 2016].

Combining (6) and (7), we propose to use

$$
\mathcal{I}_{\mathcal{F}}[\mathbf{x};\theta] \equiv \mathcal{H}[\theta] + \frac{1}{2}\mathbb{E}_{\psi(\theta)}\left[\log|2\pi e \mathcal{F}(\theta)|\right] \quad (8)
$$

as a proxy for the intractable $\mathcal{I}[\mathbf{x};\theta]$ in (5), having used $\log|\mathcal{F}^{-1}| = -\log|\mathcal{F}|$.

Note that $\mathcal{I}_{\mathcal{F}}[\mathbf{x};\theta]$ has not been proven to be a strict *bound* on $\mathcal{I}[\mathbf{x};\theta]$, but may be seen as an *approximation* to it [Wei and Stocker, 2016]. Briefly, this is because the original Stam's inequality, as stated in (7), assumes $\theta$ is a scalar location parameter, and assumes the high-precision limit where $\mathrm{q}(\hat{\theta}|\mathbf{x})$ is well-approximated by a Gaussian. Despite this, $\mathcal{I}_{\mathcal{F}}[\mathbf{x};\theta]$ is well-suited for our purposes, since (i) it leads to a remarkably simple and easy to implement expression for $\psi(\theta)$ below; (ii) we can prove that it leads to sampling when $\lambda = 1$ and VI when $\lambda \to \infty$; and (iii) we suspect that the inequality in (7) is nonetheless strict, since we neglect

the prior information contained in $\psi(\theta)$ and therefore over-estimate the conditional entropy $\mathcal{H}[\hat{\theta}|\theta]$. By analogy to the Bayesian Cramér-Rao bound [Gill and Levit, 1995, Fauß et al., 2021], a tighter variant of (7) could be derived that takes into account the prior, though possibly at the expense of added complexity; we leave this to future work.

## 4.2 CLOSED-FORM MIXING DISTRIBUTION

Substituting $\mathcal{I}_{\mathcal{F}}[\mathbf{x};\theta]$ for $\mathcal{I}[\mathbf{x};\theta]$ in (5) gives the following approximate objective,

$$
\mathcal{L}_{\mathcal{F}}(\psi,\lambda) = \mathcal{H}[\theta] + \mathbb{E}_{\psi}\left[\frac{1}{2}\log|\mathcal{F}| - \lambda\,\mathrm{KL}(\mathrm{q}||\mathrm{p}^*)\right] \quad (9)
$$

having dropped additive constants. This now resembles a maximum-entropy problem with an expected-value constraint, which has the following simple closed-form solution:

$$
\log\psi(\theta) = \frac{1}{2}\log|\mathcal{F}(\theta)| - \lambda\,\mathrm{KL}(\mathrm{q}(\mathbf{x};\theta)||\mathrm{p}^*(\mathbf{x})) \quad (10)
$$

again dropping additive constants. Equation (10) is strikingly simple, and amenable to many existing MCMC sampling methods for drawing samples of $\theta$ from $\psi(\theta)$.

Despite being derived from an approximation to our original objective, (10) nonetheless contains both sampling and VI as special cases. As $\lambda \to \infty$, the KL term dominates and $\psi(\theta)$ concentrates to $\delta(\theta - \theta^*)$, reproducing VI. When $\lambda = 1$, the resulting mixture recovers the behavior of "sampling" in the following sense:

**Definition 1 (Sampling)** *A stochastic mixture, defined by the component family* $\mathrm{q}(\mathbf{x};\theta)$ *and mixing distribution* $\psi(\theta)$, *is considered to be "sampling" if it is **unbiased** and it consists of **non-overlapping components**. An **unbiased** mixture is one where* $\mathrm{m}(\mathbf{x}) = \mathrm{p}(\mathbf{x})$. *A mixture consists of* $T$ ***non-overlapping components** if* $\sum_{t=1}^{T}\mathrm{q}(\mathbf{x};\theta_t) \approx \max_t \mathrm{q}(\mathbf{x};\theta_t)$ *with high probability.*

Lemma 4 in Appendix A.2 establishes that $\psi(\theta)$ with $\lambda = 1$ leads to sampling as defined here, assuming mixture components $\mathrm{q}$ are Gaussian. However, we conjecture that sampling arises from a broader class of $\mathrm{q}$ components as well, though computing and differentiating through $\mathcal{F}(\theta)$ for non-Gaussian component families poses additional challenges.

## 4.3 IMPLEMENTATION

Equation (10) provides a closed-form unnormalized log probability density, which is straightforward to sample from using any of a large number of existing sampling methods. For example, discrete Langevin dynamics are

$$
\theta^{(t+1)} = \underbrace{\theta^{(t)} - \gamma\lambda\nabla_\theta\mathrm{KL}(\mathrm{q}||\mathrm{p})}_{(i)} + \underbrace{\frac{\gamma}{2}\nabla_\theta\log|\mathcal{F}|}_{(ii)} + \underbrace{\sqrt{2\gamma}\eta_t}_{(iii)}
$$

where $\gamma$ is the step size and $\eta_t$ is unit isotropic Gaussian noise. This update rule is remarkably simple: $(i)$ is equivalent to gradient descent of $\mathrm{KL(q||p)}$, as done in ADVI [Kucukelbir et al., 2017], $(ii)$ biases the updates towards regions where $|\mathcal{F}(\theta)|$ is large (i.e. narrower components), and $(iii)$ adds noise.

For our experiments below, we implemented sampling from (10) in Stan [Carpenter et al., 2017], an open-source framework for probabilistic models and approximate inference algorithms. We set q to be a multivariate Gaussian with diagonal (axis-aligned) covariance, and sampled $\theta$ from $\psi(\theta)$ using Stan's default implementation of the No U-Turn Sampler (NUTS) [Hoffman and Gelman, 2014], but we emphasize that samples can be drawn from (10) using a variety of off-the-shelf sampling methods. We computed the $\mathrm{KL(q||p)}$ term using 200 random samples from q per evaluation, using the reparameterization trick to compute the gradient $\nabla_\theta \mathrm{KL(q||p)}$ and resampling the reparameterization noise only once per NUTS trajectory. This incurred a high cost in terms of number of function evaluations per sample of $\theta$, but this cost can in principle be significantly reduced by using a sampler that accepts stochastic function evaluations [Korattikara et al., 2014, Ma et al., 2015]. All comparisons to existing methods were with Stan's built-in NUTS sampler (over $\mathbf{x}$) and its built-in Automatic-Differentiation VI (ADVI) [Kucukelbir et al., 2017].

# 5 NAVIGATING BIAS/VARIANCE TRADE-OFFS FOR FINITE $T$

## 5.1 REDUCING MEAN SQUARED ERROR (MSE)

In this section, we expound the sense in which our method "interpolates" sampling and VI in terms of bias and variance. In our experiments, we quantify bias and variance in terms of the Mean Squared Error (MSE) of the expectation of an arbitrary $f(\mathbf{x})$ using a random mixture of $T$ components, $\mathrm{m}_T(\mathbf{x})$. For ADVI, we measured variance across runs with different random initializations. In Figure 3, we show empirically that by increasing $\lambda$ one can interpolate between the low bias but high variance solution, equivalent to sampling, and the low variance but high bias solution, equivalent to VI. Between these extremes, our method smoothly interpolates both bias and variance. Further implementation details can be found in Appendix B.

Computing bias and variance requires choosing a class of functions $f(\mathbf{x})$. We construct random smooth functions by discrete Fourier synthesis. Specifically, we select a series of sinusoid plane waves in the space of $\mathbf{x}$ with increasing frequency $\omega$, random directions $\mathbf{t}$ and phase $\phi$, such that $f(\mathbf{x}) = \sum_{\omega=1}^{N} a_\omega \sin(\omega \mathbf{t}^\top \mathbf{x} + \phi_\omega)$. The amplitudes $a_\omega$ are set according to a power law: $a_\omega = \omega^{-\alpha}$. Note that by choosing a random direction $\mathbf{t}$ for each frequency, it is easy to apply this definition of a random function to arbitrarily high-dimensional inference problems. An example of a 2-dimensional $f(\mathbf{x})$ is shown in Figure 3b with $\alpha = -1.5$. We vary the smoothness of the integrated function in Figure 4 by varying $\alpha$ [Stein and Shakarchi, 2011].

We also tested our algorithm on three reference problems from posteriordb [Magnusson et al., 2021], as well as on a 32-dimensional regression problem with synthetic data and known ground-truth parameters, shown in Supplemental Figure B.1. The conclusion is similar: across many random $f$s, our algorithm performs on average as well as or better than both sampling (by reducing variance) and VI (by reducing bias).

## 5.2 CONSIDERATIONS FOR SELECTING $\lambda$

A first practical consideration for the choice of $\lambda$ is the particular function $f(\mathbf{x})$ to be integrated. Since MSE can be decomposed into the sum of squared bias and variance, the value of $\lambda$ that minimizes MSE occurs when $\frac{\partial \mathrm{bias}^2}{\partial \lambda} = -\frac{\partial \mathrm{var}}{\partial \lambda}$. Any factor that increases the variance but not the bias for a fixed number of components $T$ will push the optimal $\lambda$ towards higher values (closer to VI).

One such factor is the smoothness of $f(\mathbf{x})$. Classic sampling can have problematically high variance when $f(\mathbf{x})$ is very jagged, as single points are not very representative of the surrounding function. Intuitively, then, higher $\lambda$ (more VI-like mixtures) is preferred when $f(\mathbf{x})$ is more "wiggly." To show this, we generated a random function with varying smoothness and computed their expectations them over random mixtures. The resulting MSE, bias, and variance are shown in Figure 4. We adjusted smoothness by varying the power law decay, $\alpha$, for a fixed set of phases and wave directions. At any value of $\lambda$, variance can be seen to increase as $f$ is made more wiggly ($\alpha \to -1$). With all else held equal, it is better to trade some variance for bias when the integrand changes quickly with $\mathbf{x}$.

In Figures 3 and 4, we evaluated bias and variance on the "banana" distribution. Similar results on higher-dimensional problems can be found in Supplemental Figure B.1. For illustration purposes, we chose $T$ so that the variance of NUTS was the same order of magnitude as the bias of ADVI, and estimated bias and variance by randomly subsampling sets of size $T$ from much longer chains. This approach provides theoretical insights on bias/variance trade-offs for $T$ *independent* samples, but in practice bias will be higher due to burn-in time, variance will be higher due to sampler autocorrelations, and each of these may depend nontrivially on $\lambda$. Because MCMC samplers are most effective when they have been tuned to the problem at hand, a challenge for future work will be to adapt the sampler parameters on the fly as $\lambda$ changes.

Another factor that affects the optimal $\lambda$ is the computational

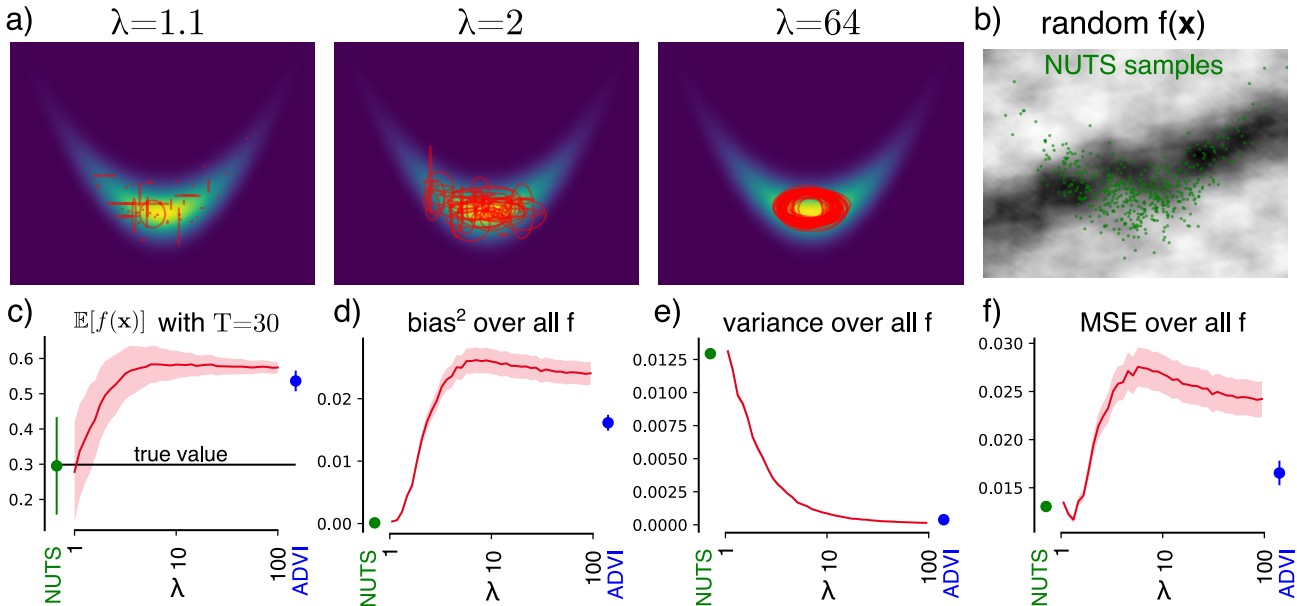

Figure 3: $\lambda$ controls a bias/variance tradeoff, interpolating between sampling and VI. **a-c)** Behavior of our method on the "banana" distribution for low, medium, and high values of $\lambda$. **b)** Example $f(\mathbf{x})$ with $\alpha = -1.5$, constructed using a random mixture of sinusoids of different frequencies and directions. Green points are values of $\mathbf{x}$ sampled using NUTS, shown for reference. **c)** The expected value of $f(\mathbf{x})$ from (b) using our method, compared with NUTS and ADVI. Error bars for NUTS and ours indicate standard deviation across runs with $T = 30$ samples each. At low $\lambda$, our method provides an unbiased but high variance estimate of $\mathbb{E}[f(\mathbf{x})]$, matching NUTS, while at high $\lambda$ it provides a bias near that of ADVI and a vanishing variance. Variance of ADVI is across 10 runs with random initializations. **d-f)** We repeated the analysis in (c) across many random $f$s (all $\alpha = -1.5$) and report the mean $\pm$ standard error of bias$^2$, variance, and MSE. MSE for our method is minimal around $\lambda \approx 1.3$ (for $T = 30, \alpha = -1.5$).

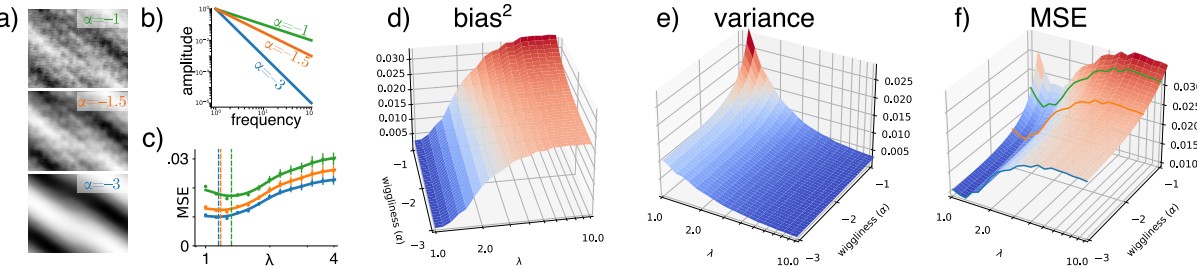

Figure 4: Interactions between $\lambda$ and integrand wiggliness for the "banana" distribution with $T = 30$. **a)** Example random integrands, $f(\mathbf{x})$, of varying degrees of wiggliness, as in Figure 3b. **b)** Wiggliness of $f$ is governed by $\alpha$, the slope of amplitude versus frequency of its component sinusoids in a log-log plot. **c)** The $\lambda$ with the smallest MSE for a fixed number samples depends on the integrand's smoothness. **d)** Bias vanishes near $\lambda = 1$. **e)** Variance is higher for smaller $\lambda$ and more wiggly integrands. **f)** MSE is the sum of bias$^2$ and variance. Overlaid lines correspond to slices shown in panel (c).

budget. In our experiments we set a fixed $T$ to demonstrate our algorithm's properties. However, if the time budget is not known in advance, a practitioner may wish to decrease $\lambda$ adaptively over time. Since variance is $\mathcal{O}(T^{-1})$, for sufficiently large $T$ error is dominated by bias, and so the optimal $\lambda$ will decay towards 1. This would result in VI-like behavior for small $T$ and sampling-like behavior for large $T$. How quickly $\lambda$ should decay will depend on the particular problem, specifically, on $\frac{\partial \text{Bias}^2}{\partial \lambda}$, and on how efficiently one can produce $T$ *independent* samples of $\theta$.

### 5.3 ANALYTICAL RESULTS

While the MSE of the expected value of some $f(\mathbf{x})$ is a useful way to compare approximate inference methods, it depends on the somewhat arbitrary choice of $f$, and in practice, the $f$'s of interest are often not known at the time of

inference. This motivates using the following alternative definition of error that is independent of $f$ and closely related to the variational objective of minimizing KL divergence:

$$\text{KL error} = \mathbb{E}[\text{KL}(\text{m}_T(\mathbf{x})||\text{p}(\mathbf{x}))] =$$
$$\underbrace{\text{KL}(\text{m}(\mathbf{x})||\text{p}(\mathbf{x}))}_{\text{KL bias}} + \underbrace{\mathbb{E}\left[\text{KL}(\text{m}_T(\mathbf{x})||\text{m}(\mathbf{x}))\right]}_{\text{KL variance}}. \quad (11)$$

That is, we define **KL bias** as the KL divergence from the infinite mixture $\text{m}(\mathbf{x})$ to the true distribution, and **KL variance** as the average KL, over realizations of $T$ independent mixture components, from $\text{m}_T(\mathbf{x})$ to the infinite mixture $\text{m}(\mathbf{x})$. Note that KL bias is identical to the infinite-mixture objective we started with in (4).

The following theorem establishes that, for all finite $T$, we can reduce the KL error relative to sampling using some $\lambda > 1$.

**Theorem 1 (Improve on sampling)** *If a mixture is sampling as in Definition 1, then $\frac{d}{d\lambda}KL\ bias = 0$ and $\frac{d}{d\lambda}KL\ variance < 0$. Thus, $\frac{d}{d\lambda}KL\ error < 0$. Proof: see Appendix A.2.*

This theorem establishes the intuitive result that the variance of sampling can be reduced, minimally impacting its bias, by replacing samples with narrow mixture components. Importantly, Theorem 1 is based on how $\psi(\theta)$ changes with $\lambda$ when using the closed-form expression for $\psi(\theta)$ we derived based on the approximate $\mathcal{L}_{\mathcal{F}}$ objective. For this theorem to apply, we must further show that both conditions of "sampling" (Definition 1) are met by $\psi(\theta)$ when $\lambda = 1$. This is proved in Lemma 4 in Appendix A.2 for Gaussian components, though we suspect it holds for other component families as well.

We can also improve on VI using our method. However, this result is slightly more subtle, as there are three cases where one should expect VI to be (locally) optimal. First, if q is in the same family as p, then $\text{q}(\mathbf{x}; \theta^*) = \text{p}(\mathbf{x})$, then is no benefit to increasing $T$, and reducing $\lambda$ only adds variance. Similarly, if q is not in the same family but the VI solution is sufficiently close to p, then a mixture of nearby qs will add variance to m [Lindsay, 1983], potentially making the match to p worse. Third, if $T$ is small – in the most extreme case, if $T = 1$ – then reducing $\lambda$ will again only add variance without reducing bias. With these three cases in mind, the following theorem establishes conditions where we expect to reduce KL error relative to VI by using a large but finite $\lambda < \infty$.

**Theorem 2 (Improve on VI)** *Assume that $\text{q}(\mathbf{x}; \theta^*)$ is poorly matched to $\text{p}(\mathbf{x})$, in the sense that $Tr\left((\nabla_\theta^2 \text{KL}(\text{q}||\text{p}))^{-1}\mathcal{F}\right) > |\theta|$, and that $\lambda$ is sufficiently large to use a Laplace approximation to $\psi(\theta)$ around $\theta^*$. Then, there exists some finite $T_0 > 1$ such that for all $T \geq T_0$, $\frac{d}{d\lambda}KL\ error > 0$. Proof: see Appendix A.3.*

Here, $|\theta|$ is the dimensionality of $\theta$. The requirement that "$\text{q}(\mathbf{x}; \theta^*)$ is poorly matched to $\text{p}(\mathbf{x})$" is expressed in terms of the curvature of $\text{KL}(\text{q}||\text{p})$ around $\theta^*$; if this curvature is small, then many "nearby" qs will also fit p well, and a mixture of them can improve on VI despite adding variance. On the other hand, the case where this curvature is not small corresponds to the earlier intuition that VI cannot be improved upon if the VI solution is already close to p. For further details, see the full proof in Appendix A.3.

# 6 DISCUSSION

**Summary:** Our work provides a new perspective on the relationship between the two dominant frameworks for approximate inference – sampling and VI – by viewing both as special cases of inference using a broader class of stochastic mixtures. Our main theoretical contribution is the framework shown in Figure 2, where mixtures that "interpolate" sampling and VI are analyzed in terms of how they trade off Mutual Information and Expected KL. We then derived an easy-to-use method based on an approximation to Mutual Information that uses the local geometry of the space of variational parameters. To demonstrate the ease and effectiveness of our method, we implemented it in the popular Stan language and demonstrated using a small set of reference problems how we "interpolate" sampling and VI by varying a single parameter, $\lambda$. Finally, we showed why such an intermediate inference scheme is useful in practice in terms of trading off bias and variance. On one hand, we proved that it is always possible to improve on classic sampling ($\lambda = 1$) by increasing $\lambda$: our method provably reduces the variance of sampling while minimally impacting its bias. Our method also provably reduces the bias of VI under certain intuitive conditions.

**Time and space complexity:** By approximating Mutual Information using only *local* geometric information in (8), in our method each component can be selected independently of the others. This means we can select and evaluate $T$ components in $\mathcal{O}(T)$ time and either $\mathcal{O}(T)$ space (if all are stored) or $\mathcal{O}(1)$ space (if components are evaluated online) – identical to traditional MCMC sampling algorithms. Further, we can run independent chains sampling $\theta \sim \psi(\theta)$ for a constant factor speedup. This improves on past work using mixture approximations, which incurred $\mathcal{O}(T^2)$ time and $\mathcal{O}(T)$ space complexity, since the optimization problem for the $T$th component depends on the location of the other $T - 1$ components, all of which must be in memory at once [Jaakkola and Jordan, 1998, Gershman et al., 2012, Zobay, 2014, Guo et al., 2016, Nalisnick and Smyth, 2017, Miller et al., 2017, Acerbi, 2018, Yin and Zhou, 2018] (but the $\mathcal{O}(T^2)$ complexity may be hardware-accelerated).

**Related Work:** The trade-offs between sampling and VI are well-studied, and many methods have been proposed to

"close the gap" between them (see [Angelino et al., 2016, Zhang et al., 2019] for general reviews). Like these other methods, we aim to provide good approximations with high computational efficiency and low variance.

There are many methods that use mixture models to reduce the bias of variational inference. Theorem 2 shows that our method only "beats" classic VI when $T > T_0$ for some finite but potentially large $T_0$. This is the price we pay for drawing mixture components stochastically [Salimans et al., 2015, Yin and Zhou, 2018]. When a mixture of $T$ components is *optimized* rather than *sampled*, bias is reduced and variance remains near zero, as in previous work [Jaakkola and Jordan, 1998, Gershman et al., 2012, Zobay, 2014, Guo et al., 2016, Miller et al., 2017], but in previous work this optimization has incurred a $\mathcal{O}(T^2)$ cost while our method is $\mathcal{O}(T)$ and can be further parallelized. Further, with some notable exceptions [Anaya-Izquierdo and Marriott, 2007, Salimans et al., 2015], most mixture VI methods make strong assumptions about the family of components [Jaakkola and Jordan, 1998, Gershman et al., 2012, Acerbi, 2018, Miller et al., 2017]. Our framework and method is somewhat agnostic to the family of q, though we have only rigorously proved that is asymptotically unbiased when using Gaussian components.

Many methods use sampling in the service of variational inference, or vice versa, but do not provide a unifying approach to both. These typically use the samples to compute expectations used to update a variational approximation [Acerbi, 2018, Miller et al., 2017, Kucukelbir et al., 2017], rather than to generate the mixture components themselves.

There is also a large number of sampling approaches that aim to improve the efficiency of sampling by reducing its variance at the cost of some bias. Some of these use variational approaches as proposal distributions, but ultimately the posterior is approximated by a set of (possibly weighted) samples of the latent variables [de Freitas et al., 2001, Korattikara et al., 2014, Ma et al., 2015, Zhang et al., 2021]. By expanding each sample from a point to a distribution, our approach allows each sample to cover more space with less variance and greater efficiency.

Despite some high-level similarities to other approaches, our framework is unusual in approximating the posterior by a sampled mixture of variational approximations. The Mixture Kalman filter [Chen and Liu, 2000] is a special case of this, which uses a sampled mixture of Gaussians, each constructed as a Kalman filter. A related approach is to *optimize* a parameterized function that generates mixture components [Salimans et al., 2015, Wolf et al., 2016, Yin and Zhou, 2018], and generative diffusion models can also be seen as a case of such mixtures [Sohl-Dickstein et al., 2015, Ho et al., 2020]. Our work differs in that we derived a closed-form mixing distribution that requires no additional learning or optimization and that is readily implemented in

existing inference software (Stan, [Carpenter et al., 2017]).

The trade-offs described in this section are summarized in Table C.1 in the Appendix.

**Limitations and future work:** Using $\mathcal{I}_{\mathcal{F}}[\mathbf{x}; \theta]$ to approximate $\mathcal{I}[\mathbf{x}; \theta]$ reduces the generality of our method, since the former is most appropriate for narrow and Gaussian-like components [Wei and Stocker, 2016]. Incorporating prior information from $\psi(\theta)$ into this bound, generalizing to other kinds of components, or even starting with alternative bounds on $\mathcal{I}[\mathbf{x}; \theta]$ are all interesting avenues for future work. Our proof of Theorem 2 requires an assumption about the curvature of KL(q||p) near the VI solution; identifying cases where this assumption holds may also be an interesting future direction.

We currently only study mixtures with $T$ *independent* mixture components without taking into account the cost of producing independent samples of $\theta$. In reality, this cost depends on the quality of the sampler, warm-up and burn-in time, and a potentially large number of calls to $\log p(\mathbf{x})$ [Zhang et al., 2021]. Further, $\lambda$ dramatically changes the shape of $\log \psi(\theta)$, which may affect the efficiency of the sampler – we mitigated this slightly by scaling the mass parameter of NUTS with $\lambda$. Other sampling algorithms besides NUTS may also be beneficial, such as ULA, which is known to have favorable scaling properties in higher dimensions [Durmus and Moulines, 2017].

We have so far considered $\lambda$ to be constant for a run of our algorithm, and this can lead to asymptotic bias even when $T$ is large. A simple adjustment to make our method effective at both small and large $T$ would be to decay $\lambda$ as $T$ grows, but note that this may require adapting the sampler parameters on the fly. Our method also requires evaluating KL(q||p) many times per sample of $\theta$. This could be made more efficient by adapting the number of Monte Carlo evaluations (fewer samples from q are sufficient when $\lambda$ is low and components are narrow), by accounting for stochastic likelihood evaluations [Ma et al., 2015], or by extending our method to mean-field message-passing [Jaakkola and Jordan, 1998], where $\nabla_\theta$KL(q||p) can be computed in closed form [Hoffman et al., 2013].

### Acknowledgements

This material is based upon work supported by the Air Force Office of Scientific Research under award number FA9550-21-1-0422, NSF CAREER IOS-1552868, and support from the McNair Foundation to XP, as well as support from the NIH R01 EY028811 to RMH. Thanks also to Daniel Lee, who acted as our guide to the Stan codebase.

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
