# OpenReview forum: "Interpolating Between Sampling and Variational Inference with Infinite Stochastic Mixtures"
_auai.org/UAI/2022/Conference — UAI 2022 Poster_

### Official Review · Reviewer_ufaN · 2022-04-09

**Q2(1) Originality/Novelty:** 2
**Q2(2) Significance/Impact:** 2
**Q2(3) Correctness/Technical Quality:** 2
**Q2(6) Clarity Of Writing:** 3
**Q6 Overall Score:** 5
**Q8 Confidence In Your Score:** 3

**Q1 Summary And Contributions:**

This paper studies the problem of sampling from an arbitrary probability distribution. The authors propose a method which interpolates between Variational Inference (VI) and sampling. The proposed method consists into performing sampling in the space of variational parameters. This extends the classical sampling and the classical VI approach leading to more flexible techniques to obtain samples from unormalized distributions. The approach relies on the expected KL/MI decomposition of the ELBO.

**Q2 Assessment Of The Paper:**

More detailed information regarding each of these aspects is given below:

**Q2(4) Quality Of Experiments (Optional):**

3: Good: The experimental evaluation is adequate, and the results convincingly support the main claims.

**Q2(5) Reproducibility:**

3: Good: Key resources (e.g., proofs, code, data) are available and key details (e.g., proofs, experimental setup) are sufficiently well-described for competent researchers to confidently reproduce the main results.

**Q3 Main Strengths:**

-The paper is very well written and easy to follow. I especially liked Figure 2 which I found to provide a clear exposition of the links between VI and sampling.

-I enjoyed the link between the bound of the ELBO found by the author and the maximum entropy approach. I think that such a decomposition could be applied to other problems and draw an important connection between maximum entropy distributions and variational inference.

-The link between VI and sampling is quite clear from the empirical point of view. Even though the experiments are low dimensional I think they successfully illustrate this point.

**Q4 Main Weakness:**

-I'm concerned about the scaling of the methods to high dimension. The number of components in a mixture to describe a high-dimensional distribution (with non-convex potential) can grow exponentially with the dimension. Therefore, I think that the method might have a limited impact in high dimensional settings. I would have appreciated a discussion and experiments regarding this issue.

-The bound they propose to estimate the Mutual Information is not a really bound but an approximation as stated by the authors. I'm wondering if this approximation makes sense in realistic settings. I understand that the authors discuss this and that in the setting where \lambda \to 1 or \lambda \to +\infty they recover the usual setting but I wonder if this inequality is verified in practice.

-Another main weakness in my point of view (which is also acknowledged by the authors) is that the computation of the Fisher matrix greatly reduce the applicability of the method (basically to Gaussian models).



**Q5 Detailed Comments To The Authors:**

-In Sec.2 I think that the link with sampling could be better explained. I feel like that the proposed method is closer to VI than it is to sampling.

-Definition 1 could be made more rigorous is it for every value of \epsilon?

-Definition 1 --> "of of"

-I think the discussion of the selection of \lambda could have been deepened. To be more precise, it would be great to have a way to automatically select \lambda depending on the target distribution. While such a result seems to be out of reach right now, I think that heuristics could be developed. This is an important problem because the method crucially relies on this \lambda parameter.

-Would it be possible to estimate \lambda while doing the sampling procedure? More precisely, would it be possible to run NUTS for k steps and then optimize the variational approximation w.r.t. \lambda for a certain number of steps (and repeat the process) in a similar fashion to [1] (or, similarly, in an Energy-Based Model fashion)?

-Theorem 2 --> "heavertailed"

-The assumptions of Theorem 2 are not precise enough. What "\lambda is large" mean? Overall I think that the statements of the theorems could be more precise

-I found the setting of Theorem 2 to be quite restrictive. Can the authors check that these assumptions are met in practice at least on toy examples?

-"Our framework and method is somewhat agnostic to the family q" (in Related works) --> I'm not sure I understand this comment because the form of the mixture distribution is very much dependent on q (see Equation 10).

-I don't see how generative diffusion models can be seen as a case of this approach. Diffusion models do not rely on an unormalized density but assume that we have access to a dataset of samples from the target distribution. This is a different setting than the one considered in this paper.

[1] -- De Bortoli, Durmus, Pereyra, Fernandez Vidal -- Efficient stochastic optimisation by unadjusted Langevin Monte Carlo

**Q7 Justification For Your Score:**

I think this is a good paper which clarifies the situation between VI and sampling. Some of the ideas introduced in the paper are interesting (I particularly enjoyed the link between maximum entropy distributions and the maximum information approximation). However, I feel that the theoretical part of the paper is a bit weak (the bound is not a bound but only an approximation, Theorems are imprecise) and I'm concerned by the applicability of the method in high dimensional settings.

**Q9 Complying With Reviewing Instructions:**

1: Yes.

---

### Official Review · Reviewer_4gT7 · 2022-04-11

**Q2(1) Originality/Novelty:** 2
**Q2(2) Significance/Impact:** 2
**Q2(3) Correctness/Technical Quality:** 3
**Q2(6) Clarity Of Writing:** 3
**Q6 Overall Score:** 3
**Q8 Confidence In Your Score:** 4

**Q1 Summary And Contributions:**

The paper presents an inference technique that tries to combine the strengths of non-parametric sampling methods with
variational inference. The approach could potentially be interesting, however there is no experimental evaluation
and comparison in high dimensional problems.


**Q2 Assessment Of The Paper:**

More detailed information regarding each of these aspects is given below:

**Q2(4) Quality Of Experiments (Optional):**

1: Poor: The experimental evaluation is flawed or the results fail to adequately support the main claims.

**Q2(5) Reproducibility:**

2: Fair: Key resources (e.g., proofs, code, data) are unavailable but key details (e.g., proof sketches, experimental setup) are sufficiently well-described for an expert to confidently reproduce the main results.

**Q3 Main Strengths:**

The approximation to the mutual information, needed in the inference objective, is interesting but it is hard to say
how will behave in challenging high-dimensional inference problems.




**Q4 Main Weakness:**

The paper is very preliminary since essentially contains (apart from few illustrative toy examples) no experiments.



**Q5 Detailed Comments To The Authors:**


I think in the next round of the work the authors need to compare against Stein variational GD
https://arxiv.org/abs/1608.04471.




**Q7 Justification For Your Score:**

Based on the current paper, it is impossible to know if the method works in practice.

**Q9 Complying With Reviewing Instructions:**

1: Yes.

---

### Official Review · Reviewer_2pK2 · 2022-04-12

**Q2(1) Originality/Novelty:** 3
**Q2(2) Significance/Impact:** 3
**Q2(3) Correctness/Technical Quality:** 3
**Q2(6) Clarity Of Writing:** 4
**Q6 Overall Score:** 6
**Q8 Confidence In Your Score:** 3

**Q1 Summary And Contributions:**

The paper presents a method for approximate inference that links the ideas of sampling and VI by modeling the approximated distribution as a mixture of "simpler“ distributions with parameters drawn from a parameterized mixing distribution. The parameter of the mixing distribution controls the sampling behavior and resembles sampling and VI in its extremes. The paper provides a theoretical contribution with a detailed derivation of the proposed method and shows its effectiveness in experiments.

**Q2 Assessment Of The Paper:**

More detailed information regarding each of these aspects is given below:

**Q2(4) Quality Of Experiments (Optional):**

2: Fair: The experimental evaluation is weak: important baselines are missing, or the results do not adequately support the main claims.

**Q2(5) Reproducibility:**

4: Excellent: Key resources (e.g., proofs, code, data) are available and key details (e.g., proof sketches, experimental setup) are comprehensively described for competent researchers to confidently and easily reproduce the main results.

**Q3 Main Strengths:**

- The paper addresses a well-known problem (representing an approximated distribution by a mixture of "simpler" distributions) and presents a novel method that improves on existing approaches in an interesting and intuitive way (e.g., it improves on runtime complexity compared to approaches that jointly optimize the mutual information).
- The submission is very well-organized and clearly written. Motivation and derivation of the presented approach are good to follow and well illustrated. There is enough information to reproduce the results (although I have not checked it myself) and the authors commit to release their implementation in STAN after publication.
- The behavior and properties of the proposed method are theoretically analyzed in detail, though only for a selected family of component distributions. Limitations of the presented approach are mentioned in several places.


**Q4 Main Weakness:**

- The effectiveness of the proposed method is demonstrated in several experiments. However, there is no comparison of its performance with related methods. The authors cite previous work in several places, for example at the beginning of section four or in section six, but do not compare their method with them empirically.

**Q5 Detailed Comments To The Authors:**

The quality of the submission could be improved by comparing the proposed method with related methods (as mentioned above). In particular, I consider the methods mentioned in the paper, which optimize the mutual information jointly. A comparison of the methods with respect to the performance depending on the computational load of each method would be insightful.

Minor remarks:
- Some internal links of the document don't seem to work properly (at least for me). I have observed this behavior especially with the links to figures and the appendix.
- In Definition 1, the word "of" appears twice consecutively.
- Figure 2: The line between the blue dot on the x-axis of the left figure (mutual information for VI) and the bottom plot of the mixture components in the center column of the figure suggests where the parameterization of the toy problem with $\lambda = 64$ is located in the left figure. However, the purple label $\lambda \rightarrow \infty$ next to the blue dot causes some confusion here, which could perhaps be resolved by repositioning the label.




**Q7 Justification For Your Score:**

Even though the experimental evaluation could be extended to also include a comparison with other related methods, the paper constitutes in my opinion a valuable contribution to the community with its detailed theoretical analysis of the proposed method and provides a basis for further research in this area. The results of the presented experiments indicate the effectiveness of the proposed method and the paper is generally well written, comprehensive and accessible.


**Q9 Complying With Reviewing Instructions:**

1: Yes.

---

### Decision · Program_Chairs · 2022-05-15

**Decision:**

Accept (Poster)

**Comment:**

Meta Review: The paper proposes an intriguing interpolation between sampling (MCMC) and variational inference. In particular, both can be seen as special cases of using an infinite mixture distribution, with different constraints on the mixing distribution. By varying the corresponding trade-off parameter, one gets interesting algorithms interpolating between these two regimes.

I have read the paper myself and found it to be of very good scholarship. The main criticism by the reviewers was lack of real-world (large-scale) inference algorithms. I can see this criticism, but find that the paper has enough merit to be accepted as is. Of course, compelling large-scale experiments would add another positive aspect and in this case I would recommend it as a top paper.

The reviewers' ratings are rather borderline and even on the negative side, but I feel that the lack of large-scale experiments got a very large weighting while the originality of the idea and good scholarship was underrated. Thus, I want to explicitly deviate from the suggested scoring and propose acceptance as a poster.

Pros:
Intriguing proposition to interpolate MCMC and VI. While some of the ideas were proposed for finite mixtures e.g. by Jaakkola and Jordan, the proposed techniques for continuous mixture distributions are original. Very good scholarship and exposition. Compelling experiments.

Cons:
Lack of large-scale experiments. Lack of comparison baselines (although I agree that the main competitors are VI and MCMC).

quality: very good

clarity: very good

originality: fair-good

significance: good